# Morphological and genomic shifts in mole-rat 'queens' increase fecundity but reduce skeletal integrity

Rachel A Johnston[1]*, Philippe Vullioud[2], Jack Thorley[2], Henry Kirveslahti[3], Leyao Shen[4], Sayan Mukherjee[3,5,6,7], Courtney M Karner[4,8], Tim Clutton-Brock[2,9], Jenny Tung[1,10,11,12]*

[1]Department of Evolutionary Anthropology, Duke University, Durham, United States; [2]Department of Zoology, University of Cambridge, Cambridge, United Kingdom; [3]Department of Statistical Science, Duke University, Durham, United States; [4]Department of Orthopaedic Surgery, Duke Orthopaedic Cellular, Developmental, and Genome Laboratories, Duke University School of Medicine, Durham, United States; [5]Department of Computer Science, Duke University, Durham, United States; [6]Department of Mathematics, Duke University, Durham, United States; [7]Department of Bioinformatics & Biostatistics, Duke University, Durham, United States; [8]Department of Cell Biology, Duke University, Durham, United States; [9]Department of Zoology and Entomology, Mammal Research Institute, University of Pretoria, Pretoria, South Africa; [10]Department of Biology, Duke University, Durham, United States; [11]Duke Population Research Institute, Duke University, Durham, United States; [12]Canadian Institute for Advanced Research, Toronto, Canada

*For correspondence:
racheljohnston7@gmail.com (RAJ);
jenny.tung@duke.edu (JT)

**Abstract** In some mammals and many social insects, highly cooperative societies are characterized by reproductive division of labor, in which breeders and nonbreeders become behaviorally and morphologically distinct. While differences in behavior and growth between breeders and nonbreeders have been extensively described, little is known of their molecular underpinnings. Here, we investigate the consequences of breeding for skeletal morphology and gene regulation in highly cooperative Damaraland mole-rats. By experimentally assigning breeding 'queen' status versus nonbreeder status to age-matched littermates, we confirm that queens experience vertebral growth that likely confers advantages to fecundity. However, they also upregulate bone resorption pathways and show reductions in femoral mass, which predicts increased vulnerability to fracture. Together, our results show that, as in eusocial insects, reproductive division of labor in mole-rats leads to gene regulatory rewiring and extensive morphological plasticity. However, in mole-rats, concentrated reproduction is also accompanied by costs to bone strength.

## Introduction

A hallmark of highly cooperative societies is reproductive division of labor. This phenomenon is best understood in eusocial insects, where environmental cues lead to reproductively and morphologically specialized castes, including one or few highly fecund 'queens' (*Wilson, 1971*). These changes help support the reproductive role of queens by differentiating them from nonbreeding colony members, who forage, care for young, and engage in colony defense (*Wilson, 1971*; *Keller and Genoud, 1997*). Queens are frequently much larger than their sterile colony mates (e.g., twice as

**eLife digest** Some social animals are highly cooperative creatures that live in tight-knit colonies. Bees and ants are perhaps the most well-known examples of social insects, while Damaraland mole-rats and naked mole-rats, two rodent species found in southern and eastern Africa, are among the most cooperative mammal species. In these colony-forming animals, only one or a few females reproduce and these fertile females are frequently referred to as "queens".

When an animal becomes a queen, her body shape can change dramatically to support the demands of high fertility and frequent reproduction. The molecular basis of such changes has been well-described in social insects. However, they are poorly understood in mammals.

To address this knowledge gap, Johnston et al. studied how transitioning to queen status affects bone growth and structural integrity in Damaraland mole-rats, as well as body shape and size. The experiments compared non-breeding female mole-rats with other adult females recently paired with a male to become the sole breeder of a new colony. Johnston et al. also used bone-derived cells grown in the laboratory to assess underlying gene regulatory changes in new queen mole-rats.

Johnston et al. showed that transitioning to the role of queen leads to a cascade of skeletal changes accompanied by shifts in the regulation of genetic pathways linked to bone growth. Queen mole-rats show accelerated growth in the spinal column of their lower back. These bones are called lumbar vertebrae and this likely allows them to have larger litters. However, queen mole-rats also lose bone growth potential in their leg bones and develop thinner thigh bones, which may increase the risk of bone fracture. Therefore, unlike highly social insects, mole-rats do not seem to have escaped the physical costs of intensive reproduction.

This work adds to our understanding of the genes and physical traits that have evolved to support cooperative behaviour in social animals, including differences between insects and mammals. It also shows, with a striking example, how an animal's genome responds to social cues to produce a diverse range of traits that reflect their designated social role.

large in honey bees and Pharaoh ants; *Berndt and Eichler, 1987*; *Page and Peng, 2001*), reflecting dramatically altered growth and development programs that are explained by changes in gene regulation (*Smith et al., 2008*). Social insects thus exemplify the tight evolutionary links between reproductive division of labor, cooperative behavior, and extreme morphological plasticity.

Systems in which breeding is restricted to a single female supported by multiple nonbreeding helpers are also observed in vertebrates, including birds and mammals (*Koenig and Dickinson, 2016*). Here, breeding status is not determined during early development, but instead occurs in adulthood, and breeding is only achieved by those individuals who have the opportunity to transition into a reproductive role. In some species, new breeders undergo a period of accelerated growth, which may be important either for maintaining dominance or for supporting high fecundity (*Clutton-Brock et al., 2006*; *Huchard et al., 2016*; *O'Riain et al., 2000*; *Russell et al., 2004*; *Thorley et al., 2018*; *Young and Bennett, 2010*). While substantial gene regulatory divergence with breeding status has been described for the brain and some peripheral organs (*Bens et al., 2018*; *Mulugeta et al., 2017*; *Sahm et al., 2020*), we know little about the gene regulatory shifts responsible for breeder-associated patterns in growth. Because morphological change is often crucial for ramping up offspring production, these processes are key to understanding both the basis for, and limits of, status-driven differences in growth and development.

Here, we investigate the morphological and molecular consequences of experimental transitions to breeding status in female Damaraland mole-rats (*Fukomys damarensis*). Like naked mole-rats (*Heterocephalus glaber*), Damaraland mole-rats are frequently classified as 'eusocial' (*Bennett and Faulkes, 2000*; *Jarvis, 1981*; *Jarvis and Bennett, 1993*), and female helpers who transition to queens experience accelerated vertebral growth associated with increases in fecundity (*O'Riain et al., 2000*; *Thorley et al., 2018*). However, it is not clear what triggers skeletal remodeling, where it is localized within the vertebral column, or whether it extends to other parts of the skeleton. Further, the gene regulatory changes that support skeletal remodeling in mole-rat queens are not known, nor are their consequences for skeletal growth potential and integrity. To address these questions, we experimentally assigned age-matched, female littermates to become queens or

remain as nonbreeders and evaluated gene regulatory and morphological changes induced by the transition to queen status. Our results indicate that, as in eusocial insects, females that acquire breeding status experience substantial morphological remodeling, associated with pathway-specific changes in gene regulation. Notably, we found that queens not only experience lengthening of their lumbar vertebrae (LV), but also show reductions in the growth potential and structural integrity of their long bones. These changes result from increased rates of bone resorption that may increase the risk of fracture, indicating that the presence of helpers does not annul the costs of reproduction to queens.

## Results

### Adaptive plasticity in the skeleton of Damaraland mole-rat queens

Adult female Damaraland mole-rats were randomly assigned to either transition to queen status (n = 12) or remain as nonbreeders (n = 18) for the duration of the experiment (*Figure 1A–C*; *Supplementary file 1*). Age at assignment (mean age = 19.4 ± 4.4 s.d. months) was consistent with the age at dispersal observed in wild Damaraland mole-rats (1–3 years, Thorley and Clutton-Brock, unpublished data). To resolve whether skeletal changes are a function of the queen transition per se versus release from reproductive suppression in the natal colony, nonbreeders were either kept in their natal colonies as helpers or placed into solitary housing in the absence of a breeding queen, recapitulating extended periods of dispersal in this species (*Jarvis and Bennett, 1993*) (n = 10 helpers and n = 8 solitaires). At the time of assignment, females assigned to the queen, helper, and solitaire treatments were statistically indistinguishable in body mass, age, and vertebral length (as measured by LV5; unpaired t-tests between all pairwise combinations of treatments: p>0.05; *Figure 1—figure supplement 1*). When possible, we assigned age-matched littermates to queen versus nonbreeding treatments (26 of 30 experimental animals were in sets of littermate sisters; *Supplementary file 1*). Six nonexperimental animals (one queen and five nonbreeders) were also included in the sample, resulting in a total sample size of 13 breeders and 23 nonbreeders (*Supplementary file 1*).

Females assigned to the queen treatment were each transferred to a new tunnel system containing only an unrelated adult male, simulating the natural process of dispersal and new colony formation in the wild (*Jarvis and Bennett, 1993*). This pairing procedure, which defines the queen treatment, typically leads to immediate sexual activity and rapid activation of the reproductive axis, including initiation of ovulation and the potential for conception (*Bennett et al., 1996*; *Snyman et al., 2006*). Queens gave birth to a mean of 6.92 ± 5.57 s.d. live offspring during the 12–22-month follow-up period, produced in a mean of 2.85 ± 1.75 s.d. litters (range: 0–6; *Supplementary file 1*). As expected, helpers and solitaires produced no offspring, and did not differ from each other in body mass or vertebral length after the 12–22-month follow-up period (unpaired t-tests, all p>0.05; *Figure 1—figure supplement 2*). Because helpers and solitaires were morphologically indistinguishable, and also exhibited no differences in gene expression in our subsequent genomic assays (*Supplementary file 2*), we grouped them together into a single 'nonbreeder' treatment for the remainder of our analyses.

Compared to nonbreeders, queens showed rapid growth in the LV in the first 12 months post-pairing (*Figure 1D, E*), especially in the vertebrae toward the caudal end of the vertebral column (LV5 and LV6). Based on longitudinal measurements, most of this differential growth was concentrated soon after the breeding status transition. Specifically, we observed a significant interaction between breeding status (queen versus nonbreeder) and post-pairing time point in the first four months of the experiment (*Figure 1D*; β = 0.0794, p=3.13×10$^{-3}$; n = 49 x-rays from 28 animals), but not for measurements taken in later time point intervals (4 versus 8 months; 8 versus 12 months, 12 versus 16 months, 16 versus 22 months; all p>0.05). Moreover, in the first four months, only queens that had already experienced pregnancy showed accelerated vertebral lengthening relative to nonbreeders (unpaired t-test; LV5 of pregnant queens versus nonbreeders: t = −5.735, df = 16.871, p=2.50×10$^{-5}$; LV5 of queens not yet pregnant versus nonbreeders: t = −0.789, df = 13.007, p=0.444; n = 14 nonbreeders, five pregnant queens, and two queens not yet pregnant).

As a result of accelerated vertebral growth in queens post-transition, size differences persisted throughout the study. After 12 months, the absolute length of LV5 in queens was, on average, 4.8%

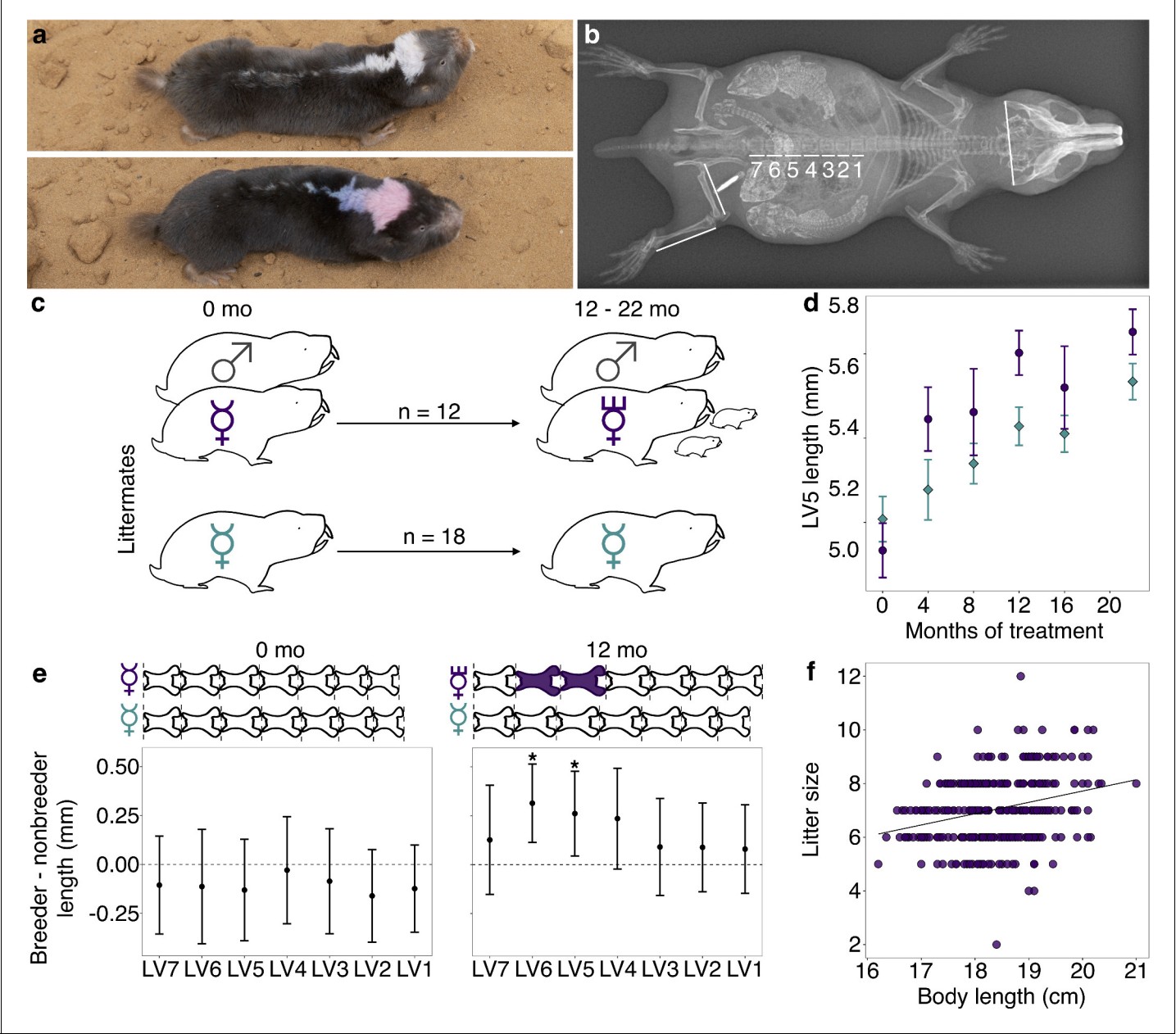

**Figure 1.** Transition to queen status leads to lumbar vertebral (LV) lengthening. (**a**) A breeding 'queen' (top) and nonbreeding female (bottom) Damaraland mole-rat, shown at the same scale. Animals are dye marked to allow for individual identification. Photo credit James Bird. (**b**) An x-ray of a female breeder, with lines depicting the x-ray measurements taken (LV1–LV7, right femur, right tibia, and zygomatic arch). Three developing offspring are visible within the abdomen (highlighted by increased brightness and contrast), and span the length of the LV. (**c**) Experimental design: nonbreeding adult female ( ☿ ) littermates were randomly assigned to transition to queen status (purple ☿ ) by being paired with an unrelated male ( ♂ ), or to remain in a nonbreeding treatment (cyan). Duration of treatment ranged from 12 to 22 months. (**d**) Queens (purple dots) show more rapid growth in LV5 in the first four months of the experiment, relative to nonbreeders (cyan diamonds; treatment by time point interaction: β = 0.078, n = 49, p=3.47×10$^{-3}$). Dots show means ± standard errors (bars). (**e**) At the start of the experiment (0 months, left panel), the LV of breeders do not differ from those of nonbreeders (unpaired t-tests, all p>0.05). However, at 12 months (right panel), queens have longer LV relative to nonbreeders (unpaired t-tests, * indicates p<0.05). Dots show means ± standard errors (bars). Lengths of LV above the plots are scaled to indicate the mean lengths of queens (top) and nonbreeders (bottom) at each time point; vertebrae highlighted in purple are significantly longer in queens relative to nonbreeders. (**f**) Litter size is positively correlated with maternal body length in the Damaraland mole-rat colony (β = 0.353, n = 328 litters, p=1.35×10$^{-3}$).

The online version of this article includes the following figure supplement(s) for figure 1:

**Figure supplement 1.** Mass, age, and lumbar vertebra (LV) 5 length of queen, helper, and solitaire female mole-rats at the start of the experiment.

*Figure 1 continued on next page*

*Figure 1 continued*

**Figure supplement 2.** Mass and lumbar vertebra (LV) 5 length of helper and solitaire female mole-rats after 12–22 months of experimental treatment of social status.

**Figure supplement 3.** Body length is positively correlated with the lengths of lumbar vertebrae (LV) 1–7.

longer than nonbreeders (*Figure 1E*; LV5: unpaired t-test, t = 2.509, df = 21.095, p=0.020), and the absolute length of the LV column in queens relative to nonbreeders was 3.5% longer, although the latter difference was not significant (unpaired t-test, t = 1.945, df = 22.49, p=0.064). Differences between queens and nonbreeders were even more apparent if LV measures were scaled by zygomatic arch (head) width, as in previous studies (*O'Riain et al., 2000*; *Thorley et al., 2018*; *Dengler-Crish and Catania, 2009*) (LV5: 9.3% longer, unpaired t-test, t = 4.12, df = 15.135, p=$8.87\times10^{-4}$; LV column length: 7.9% longer, unpaired t-test, t = 4.34, df = 15.37, p=$5.58\times10^{-4}$). Thus, transitions to queen status induce reproductive investment, which in turn leads to organism-wide allometric changes that generate an elongated phenotype.

The elongated phenotype appears to subsequently facilitate future fecundity. Queens with longer bodies (which correlates with longer LV, Pearson's *r* = 0.856, p=$5.99\times10^{-59}$; *Figure 1—figure supplement 3*) had more pups per litter (*Figure 1F*; β = 0.353, p=$1.35\times10^{-3}$, n = 328 litters from all breeding groups maintained in the same breeding facility; *Supplementary file 3*). Controlling for litter size, longer queens also had larger pups: for every additional centimeter of maternal body length, pup body mass increased by 2.9% (β = 0.29, p=0.032, n = 971 pups). Thus, the elongated queen phenotype is a strong candidate for adaptive plasticity that supports increased fertility in queen mole-rats.

## Breeding status induces gene regulatory changes in the queen mole-rat skeleton

To identify the gene regulatory changes associated with skeletal plasticity, we cultured cells enriched for bone marrow-derived mesenchymal stromal cells (bMSCs) isolated from the LV (pooled LV1–LV5) of queens and nonbreeders (n = 5 queens, 11 nonbreeders). bMSC cultures include multipotent skeletal stem cells, the precursor of the osteoblast and chondrocyte lineages responsible for bone growth. In parallel, we cultured cells enriched for bMSCs from the pooled long bones (humerus, ulna, radius, left femur, and left tibia) of the same animals, which do not show increased elongation in queens (femur at 12 months: unpaired t-test, t = −0.202, df = 19.326, p=0.842; tibia at 12 months: unpaired t-test, t = −0.860, df = 16.759, p=0.402). To evaluate the potential role of sex steroid hormone signaling on bone growth, we treated cells from each bone sample for 24 hr with either 10 nM estradiol or vehicle control, resulting in 47 total samples. We then performed RNA-Seq on each sample to screen for genes that were systematically differentially expressed in the bone cells of queens versus nonbreeders.

Of 10,817 detectably expressed genes, 171 genes showed a significant effect of breeding status at a false discovery rate (FDR) threshold of 10% in the long bones (329 at an FDR of 20%; *Supplementary file 4*). Surprisingly, no genes showed a significant effect of breeding status in the LV at either FDR threshold. However, effect sizes were highly correlated between bone types overall ($R^2$ = 0.75, p=$4.60\times10^{-53}$), with more pronounced effects of breeding status in the long bone samples than in the LV (paired t-test on breeding status effects in long bone versus vertebrae: t = 3.97, df = 317.67, p=$8.73\times10^{-5}$). Importantly, breeding status-related differences were not readily attributable to differences in bone cell composition. Based on both canonical markers of bMSC lineage cells and deconvolution of the RNA-Seq data using data from 27 mesenchymal or hematopoietic lineage mouse cell types, the majority cell type in both queen and nonbreeder samples was most similar to cells from the bMSC lineage (*Dominici et al., 2006*; *Hume et al., 2010*; *Newman et al., 2015*; *Figure 2—figure supplements 1* and *2*). Additionally, the top three principal components summarizing estimated cell-type proportions did not differ between queens and nonbreeders (all FDR > 10%, *Supplementary file 5*), and we identified no cases in which the effects of breeding status on gene expression were significantly mediated by the first principal component of cell composition (p>0.05 for all 171 queen-associated genes at 10% FDR; *Supplementary file 6*).

The majority of breeding status-associated genes were upregulated in queens (151 of 171 genes, 88%). In support of their role in skeletal plasticity, upregulated genes were enriched for bone remodeling ($\log_2[OR]$=4.07, p=$5.07\times10^{-6}$), a process that involves the balanced cycle between bone formation by osteoblasts and bone resorption by osteoclasts (*Redlich and Smolen, 2012*; *Figure 2*). Surprisingly, however, enriched pathways were specifically related to bone resorption, not formation (*Supplementary file 7*), including 'positive regulation of bone resorption' (*Figure 2A, C*; $\log_2[OR]$ =6.51, p=$1.55\times10^{-6}$) and 'superoxide anion generation,' which is involved in osteoclast activity and degradation of bone matrix (*Figure 2A, C*; $\log_2[OR]$=5.29, p=$1.4\times10^{-5}$) (*Darden et al., 1996*; *Datta et al., 1996*; *Key et al., 1990*; *Key et al., 1994*). Differentially expressed genes were also

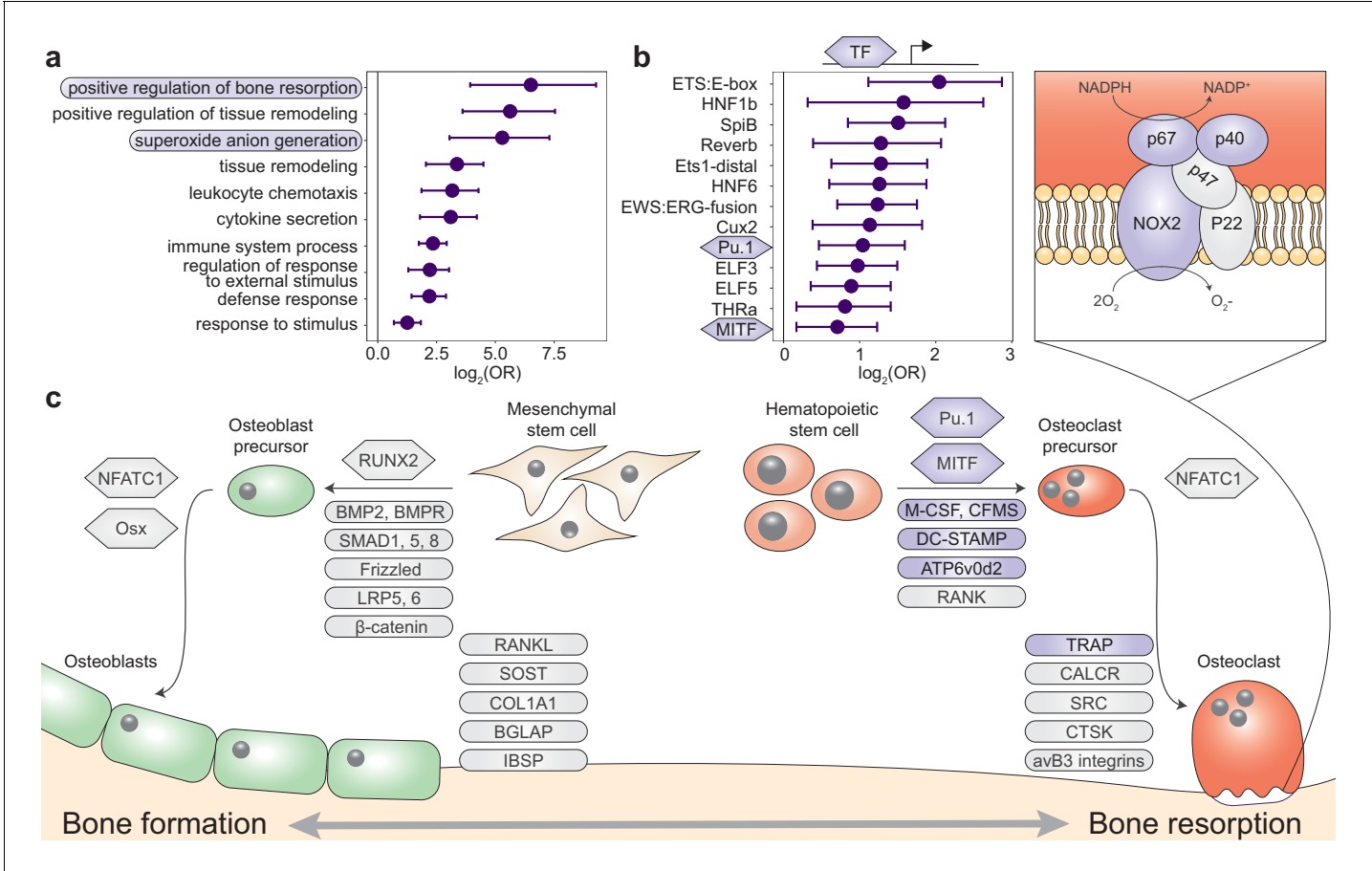

**Figure 2.** Queen status drives increased regulatory activity of bone resorption pathways. (a) Gene Ontology (GO) terms enriched in queen upregulated genes, relative to the background set of all genes tested. Bars represent 95% confidence intervals. Processes highlighted in purple are also depicted in (c). Highest-level (most general) terms are shown; for full GO enrichment results, see *Supplementary file 7*. (b) Accessible transcription factor binding site motifs enriched near queen upregulated genes, relative to all genes tested. Bars represent 95% confidence intervals. Transcription factors highlighted in purple are also depicted in (c). (c) Schematic of the balance between bone formation and bone resorption, showing key regulators and markers for mesenchymal stem cell differentiation into osteoblasts and hematopoietic stem cell differentiation into osteoclasts (*Redlich and Smolen, 2012*; *Segeletz and Hoflack, 2016*). Note that not all genes or proteins in gray were tested for differential expression (e.g., because they were not annotated in the Damaraland mole-rat genome or were too lowly expressed in our sample); see *Supplementary file 4* for full set of tested genes. Queen upregulated genes or corresponding proteins (false discovery rate [FDR] < 10%) are highlighted as purple ovals, and transcription factors with binding motifs enriched near queen upregulated genes are highlighted as purple hexagons. Inset for osteoclasts shows the NADPH oxidase system, which generates superoxide radicals ($O_2^-$) necessary for bone resorption and is highly enriched for queen upregulated genes (purple ovals).

The online version of this article includes the following figure supplement(s) for figure 2:

**Figure supplement 1.** Mole-rat RNA-Seq samples cluster closest with purified mouse osteoblasts, based on canonical bone marrow-derived mesenchymal stromal cell (bMSC) markers.

**Figure supplement 2.** Estimated cell proportions for the 47 mole-rat RNA-Seq samples.

**Figure supplement 3.** Footprint profiles of transcription factors androgen receptor (AR), estrogen receptor 1 (ESR1), and estrogen receptor 2 (ESR2).

enriched for immune-related processes (e.g., 'cytokine secretion,' 'chemotaxis,' 'leukocyte activation involved in immune response'; *Supplementary file 7*). These observations suggest that transitions to queen status also involve changes in immunoregulatory signaling (osteoclast cells are derived from monocytes).

Omni-ATAC-seq profiling of open chromatin regions further supports a central role for bone resorption and osteoclast activity in the queen skeleton (n = 8; *Supplementary file 8*). Specifically, transcription factor binding motifs (TFBMs) located in accessible chromatin near queen upregulated genes were enriched for PU.1 and MITF, two transcription factors that are essential for osteoclast differentiation (*Segeletz and Hoflack, 2016*; *Figure 2B, C*; PU.1 $\log_2[OR]=1.041$, $p=2.84\times10^{-4}$; MITF $\log_2[OR]=0.707$, $p=7.36\times10^{-3}$; see *Supplementary file 8* for a complete list of enriched TFBMs). *MITF* was also among the 151 genes that were differentially expressed between queens and nonbreeders and upregulated in both queen long bones and LV. Surprisingly, given the role of sex steroid hormones in bone growth and elevated estradiol levels in queen versus helper Damaraland mole-rats (*Bennett, 1994*), we observed no significant effects of estradiol treatment on gene expression in either bone type (all FDR > 10%). Queen upregulated genes were also not in closer proximity to androgen response elements (AREs) or estrogen response elements (EREs) than expected by chance (ARE $\log_2[OR]=0.207$, $p=0.627$; ERE $\log_2[OR]=0.196$, $p=0.652$). Consistent with this observation, transcription factor footprinting analysis showed no evidence of queen-associated differences in transcription factor activity of the androgen receptor (AR), estrogen receptor 1 (ESR1), or estrogen receptor 2 (ESR2), in either the long bones or LV (all paired t-tests: p>0.05; *Figure 2— figure supplement 3*). Thus, our data point to the involvement of non-sex steroid-mediated signaling pathways in remodeling queen mole-rat bones, at least after 1 year post-transition.

## Extensive skeletal remodeling in queen mole-rats

The gene expression data suggest that queen status-driven changes to the skeleton extend beyond the LV to the long bones. Further, they suggest that bone resorption—an important counterpoint to bone formation that is required for normal skeletal maintenance—also distinguishes breeding and nonbreeding females. To investigate this possibility, we performed high-resolution micro-computed tomography (μCT) scanning to generate 3D reconstructions of LV6, LV7, right femur, and right tibia of queens and female nonbreeders (n = 140 bones from 36 animals; *Figure 3A*, *Figure 3—figure supplement 1*). This approach substantially increases the level of resolution for investigating breeding status-linked differences in skeletal morphology, as previous studies relied on x-ray data alone (*O'Riain et al., 2000*; *Thorley et al., 2018*).

We first asked whether breeding status could be predicted from morphological differences in the 3D reconstructions. We found that it could for the LV, but not for the femur: by applying the smooth Euler characteristic transform (*Crawford et al., 2016*), we were able to predict queen versus nonbreeder status in LV6 (77.8% accuracy, p=0.01, n = 36), but not the femur (52.8% accuracy, p=0.53, n = 36). Including only highly fecund queens (≥6 total offspring) improved predictive accuracy in the femur (70% accuracy, p=0.12, n = 30). Although these predictions did not reach statistical significance, they raised the possibility that morphological changes in femurs become enhanced with increasing reproductive effort.

We next tested whether the transition to queen status affects the ability to continue bone lengthening. Lengthening requires the presence of a growth plate, a region of cartilage in the bone where longitudinal growth occurs through proliferation of cartilage cells (chondrocytes) (*Figure 3A, B*, *Figure 3—figure supplement 1*). Closure of the growth plate, which indicates that bone lengthening potential has terminated, typically occurs in mammals after reaching sexual maturation, when energy begins to be invested in reproduction instead of growth (*Kilborn et al., 2002*). To test whether the transition to queen status alters bone lengthening potential, we performed Safranin O staining on sections of the right tibia and LV7 to visualize growth plates (*Figure 3B*). In the proximal tibia but not LV7, queens were less likely to have open growth plates (*Figure 3C*, *Figure 3—figure supplement 1*, and *Supplementary file 9*; tibia: two-sided binomial test, p=0.019; LV7: two-sided binomial test, p=0.422). The increased probability of growth plate closure in the tibia of queens is linked to the number of offspring a female has produced: females with more offspring showed a higher expanse of closure across the growth plate (β = 0.050, $p=4.51\times10^{-3}$, n = 12, controlling for age). This pattern may be due in part to reduced chondrocyte proliferation, as females that produced more offspring had fewer chondrocyte columns in the remaining growth plate (*Figure 3D*;

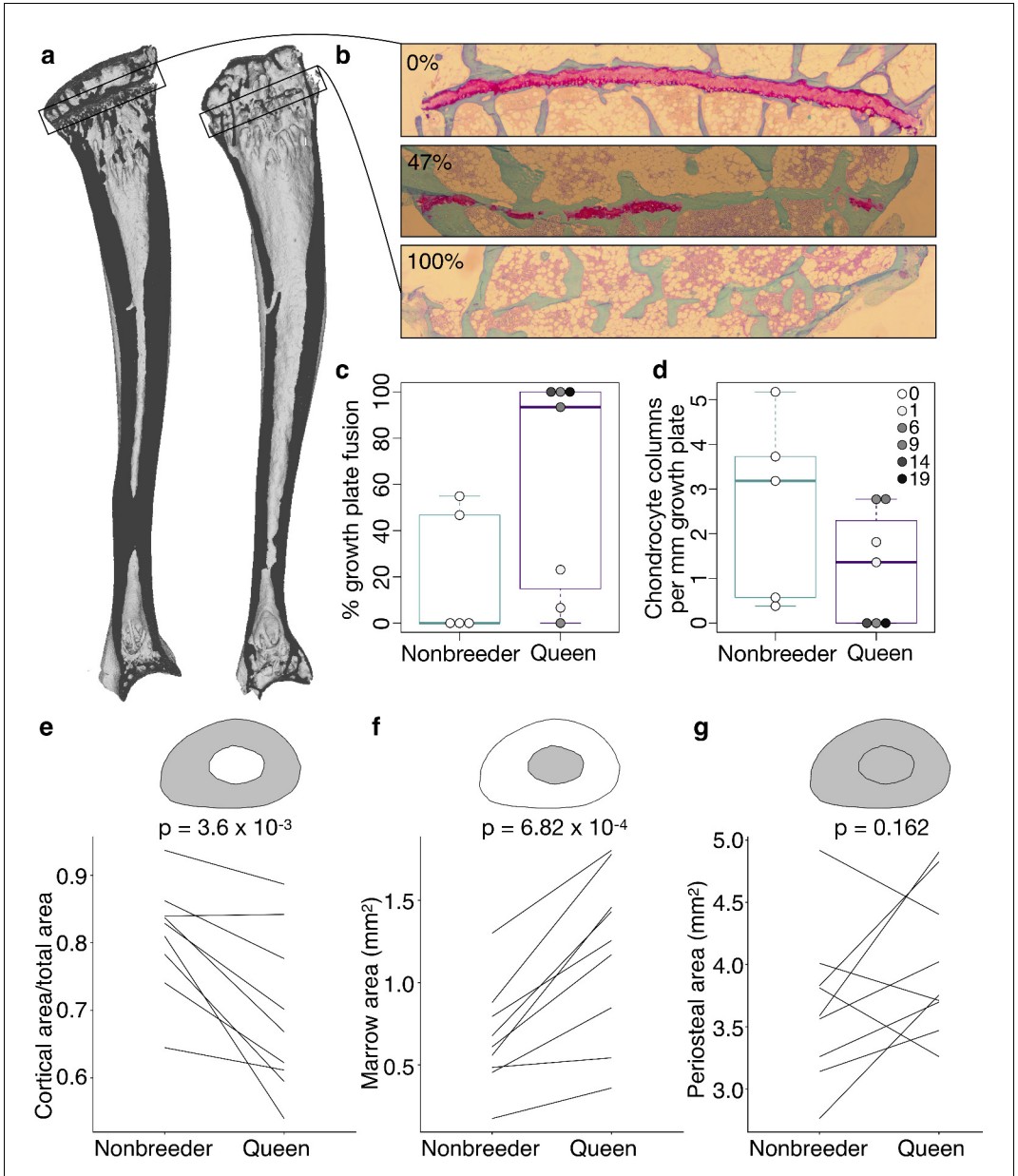

**Figure 3.** Queen status leads to reduced growth potential in the tibia and reduced cortical area at the femoral midshaft. (a) Micro-computed tomography (μCT) scans of Damaraland mole-rat tibias. Boxes indicate the location of the proximal growth plate, which varies between unfused (left) to fully fused (right). (b) Example Safranin O-stained histological sections of the proximal tibia, in which the growth plate is unfused (top), partially fused (middle), or fully fused (bottom). Values indicate percent growth plate fusion across the width of the bone. The cartilaginous growth plate is stained deep pink, and calcified bone is stained green. (c) Queens, and specifically queens that gave birth to more offspring, show increased growth plate fusion ($\beta$ = 0.050, p=4.51×10$^{-3}$, n = 12, controlling for age) and (d) decreased number of chondrocyte columns within the remaining growth plate ($\beta$ = −0.132, p=0.020, n = 12, controlling for age). Each box represents the interquartile range, with the median value depicted as a horizontal bar. Whiskers extend to the most extreme values within 1.5× of the interquartile range. In (c) and (d), dots represent individual animals, and shading indicates each animal's total offspring number. Ages of queens and nonbreeders do not significantly differ (unpaired t-test, t = 0.489, n = 12, p=0.644). (e–g) Femoral cross-sections with area highlighted in gray show the measures represented in the corresponding plots below. Each line represents an age-matched, nonbreeder and queen littermate pair. (e) Queens have less cortical bone (relative to the total area of the femoral midshaft cross-section) compared to their paired nonbreeding littermates (paired t-test, t = −4.07, df = 8, p=3.60×10$^{-3}$). (f) Queens also have enlarged marrow cavities (paired t-test, t = 5.36, df = 8, p=6.82×10$^{-4}$) but (g) show no difference in overall periosteal area (paired t-test, t = 1.54, df = 8, p=0.162). The online version of this article includes the following figure supplement(s) for figure 3:

**Figure supplement 1.** Growth potential in lumbar vertebra (LV) 7.

**Figure supplement 2.** Queens do not exhibit reduced cortical area at the midsection of lumbar vertebra (LV) 6.

β = −0.132, p=0.020, n = 12, controlling for age). Thus, offspring production in queens is associated with loss of the ability to lengthen the long bones, but not the LV, consistent with the importance of abdominal lengthening for supporting larger litters.

A major demand on reproductively active female mammals is a high requirement for calcium, particularly during lactation when mothers support rapid offspring bone growth. Maternal skeletons are remodeled to meet this demand, although in most mammals, these changes are not permanent (reviewed in *Kovacs, 2016*). Because of the particularly intense reproductive investment made by cooperatively breeding mole-rat queens, we therefore also evaluated the effect of queen status on trabecular and cortical bone volumes, which are thought to be important in satisfying short-term and long-term calcium demands, respectively. We found no effect of queen status on the amount of trabecular bone in the femur, tibia, LV6, or LV7 (all p>0.05 for bone volume/total volume). However, we found that cortical bone was significantly reduced at the femoral midshaft, but not in the LV, in queens compared to their nonbreeding sisters (*Figure 3E*, *Figure 3—figure supplement 2*; femur: paired t-test of cortical area (CA)/total area, t = −4.067, df = 8, p=3.60×10⁻³; LV6: paired t-test of CA/total area, t = −0.741, df = 6, p=0.487). Relative to nonbreeders, the femoral midshafts of queens showed significantly lower apparent density (paired t-test, t = −3.734, df = 8, p=5.75×10⁻³), but no difference in material density (paired t-test, t = −0.074, df = 8, p=0.943). Thus, reduced bone mass in queens is due to reduced bone volume, and not to reduced mass per unit volume (e.g., due to increased porosity).

The reduction of apparent density in queens could be due to slowed bone growth, which typically occurs on the outer, periosteal bone surface, or to increased bone resorption, which typically affects the inner endosteal surface and increases the marrow cavity. Our analysis indicates that cortical bone loss in queens is due to the latter explanation: queens had a larger marrow cavity (paired t-test, t = 5.355, df = 8, p=6.82×10⁻⁴; *Figure 3F*) but showed no difference in periosteal area compared to their nonbreeding sisters (paired t-test, t = 1.539, df = 8, p=0.162; *Figure 3G*).

Because changes in cortical bone are thought to reflect accumulated demands over long time frames, we hypothesized that cortical thinning in queens is a consequence of repeated cycles of pregnancy and lactation over time, which can occur simultaneously in Damaraland mole-rat queens. In support of this idea, we found that, within queens, the relative amount of cortical bone is not predicted by the number of pups in a queen's recent litter (pups born within the past 30 days; β = −0.024, n = 13, p=0.287), but instead by the total number of pups she produced in her lifetime. Specifically, queens who had more live births had reduced cortical bone thickness along the entire shaft of the femur (*Figure 4* and *Supplementary file 10*; across decile sections of the femur: all p<0.05, controlling for mother's litter as a random effect). Thus, cortical thinning does not commence with the transition to queen status per se (i.e., it is not a correlate of *achieving* breeder status), but instead appears to be a consequence of repeated investment in pregnancy and lactation. Notably, thinning is particularly marked in queens who had at least six offspring, which usually occurs by 14 months after a breeding status transition (i.e., by the third litter; *Supplementary file 10*). Given that wild Damaraland mole-rat queens can maintain their status for many years (*Schmidt et al., 2013*), our results suggest that long-lived queens may experience substantial morphological change (although the long lives of wild Damaraland mole-rat queens [*Schmidt et al., 2013*] suggest these changes must be manageable to some degree, or potentially even recoverable).

## Skeletal remodeling predicts increased risk of femur breakage in queens

In humans, accelerated bone resorption is a central cause of osteoporosis-related bone fragility (*Szulc et al., 2006*). We therefore hypothesized that cortical thinning in queen mole-rat femurs would be linked to decreased bone strength. To test this hypothesis, we calculated three key indicators of femoral structural integrity: cortical area (CA), the minimum second moment of inertia ($I_{min}$, a correlate of minimum resistance to bending), and polar second moment of area (J, a correlate of torsional rigidity). In nonbreeders, the three measures are positively correlated with body mass (CA: $R^2$ = 0.409, n = 21, p=1.08×10⁻³; $I_{min}$: $R^2$ = 0.412, n = 21, p=1.03×10⁻³; J: $R^2$ = 0.422, n = 21, p=8.62×10⁻⁴). However, in queens, $I_{min}$ and J are not significantly predicted by body mass ($I_{min}$: $R^2$ = 0.088, n = 13, p=0.17; J: $R^2$ = 0.239, n = 13, p=0.0514), but are instead a function of number of offspring produced ($I_{min}$: $R^2$ = 0.283, p=0.0354; J: $R^2$ = 0.271, n = 13, p=0.0393). Queen CA is

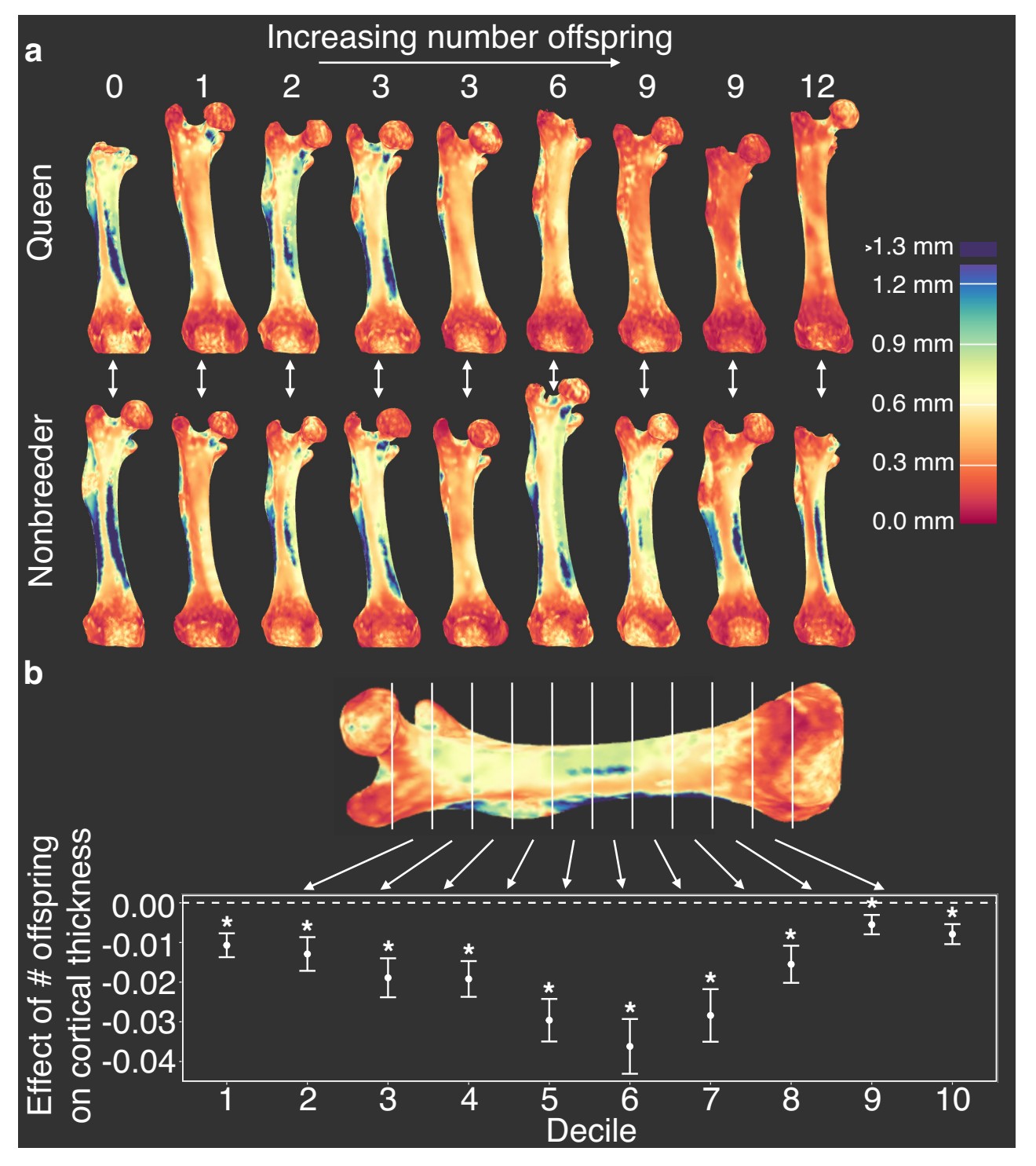

**Figure 4.** Offspring production in queens leads to cortical thinning across the femoral shaft. (**a**) Queens (top row) relative to their same-aged female nonbreeding littermates (bottom row) present thinner cortical bone across the femur, particularly in females that have many offspring (top right). Number of offspring is indicated above each femur, and vertical arrows indicate littermate pairs. (**b**) Within each decile section across the femoral shaft, number of offspring is negatively correlated with average cortical thickness (linear mixed model with littermate pair as random effect). Full results are presented in *Supplementary file 10*. Asterisk indicates p<0.05.

predicted by both offspring number and body mass, but offspring number explains almost twice the variance (offspring number $R^2$ = 0.634, p=6.83×10$^{-4}$; body mass $R^2$ = 0.385, p=0.014).

To evaluate the effects of reproductive activity on the risk of bone failure, we drew on data on the relationship between CA and bone mechanical failure in a large data set of mouse femurs (*Jepsen et al., 2003*). In this data set, CA is the best predictor of maximum load (the maximum force a bone can withstand prior to failure), and, crucially, the CA-max load relationship is highly linear (*Figure 5—figure supplement 1*; $R^2$ = 0.877, p=6.64×10$^{-38}$). Scaling the mole-rat CA data to mouse suggests that transitions to queen significantly increase the risk of bone failure (*Figure 5*; hazard ratio [95% confidence interval]=2.68 (1.18, 6.08), n = 34, p=0.018). Similar to growth potential and cortical thinning, this effect is driven by highly fertile queens, such that those who had at least six offspring showed the highest predicted risk of bone failure (*Figure 5*; queens with ≥6 offspring relative to nonbreeders: HR = 3.74 (1.42, 9.82), n = 28, p=0.007). The risk of bone failure is thus predicted to increase by 21% for each additional pup (HR = 1.21 (1.10, 1.33), n = 34, p=1.22×10$^{-4}$).

## Discussion

Our results demonstrate that transitions to breeding status in Damaraland mole-rat queens lead to a cascade of skeletal changes linked to shifts in gene regulation. The vertebral lengthening observed in Damaraland mole-rat queens is concordant with previous reports of vertebral lengthening in both Damaraland mole-rats (*Thorley et al., 2018*) and naked mole-rats (*O'Riain et al., 2000*). Like naked mole-rats, our analyses show that most growth is concentrated soon after the breeding status transition, especially in connection with the first post-transition pregnancies (*Dengler-Crish and Catania, 2009*; *Henry et al., 2007*). However, our findings also suggest subtle differences: for instance, while the growth phenotype in naked mole-rats occurs at the cranial end of the LV (*Henry et al., 2007*), it is concentrated at the caudal end of the vertebral column in Damaraland mole-rats. Given that Damaraland mole-rats and naked mole-rats independently evolved a similar, highly cooperative social structure (*Jarvis and Bennett, 1993*; *Faulkes and Bennett, 2016*), this difference suggests potential convergent evolution of the vertebral lengthening phenotype in queens, presumably in response to the selection pressure for increased fertility.

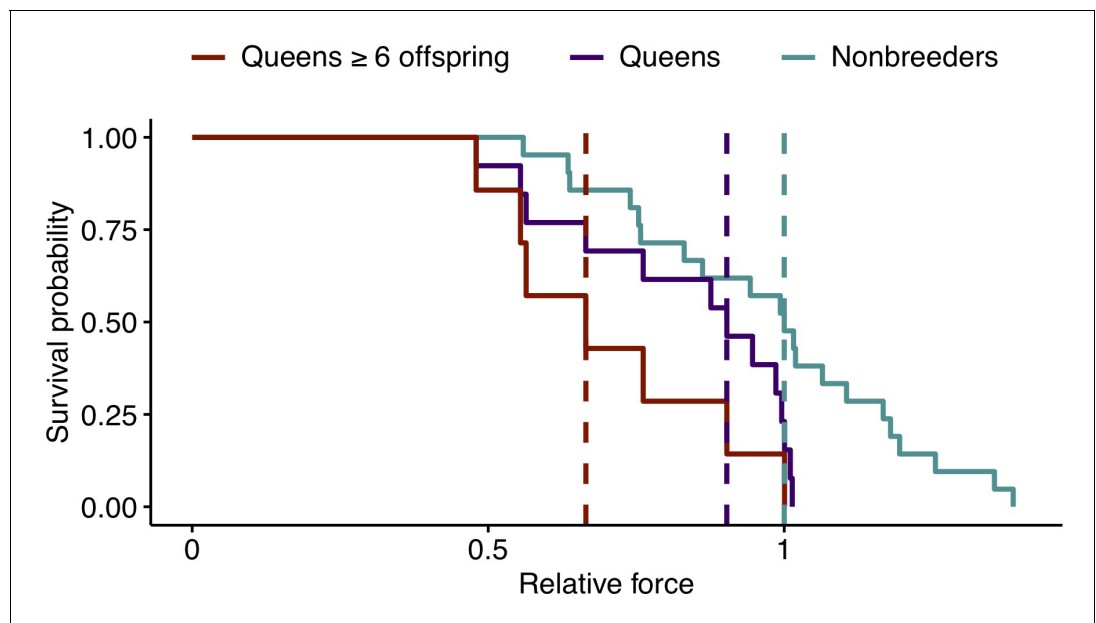

**Figure 5.** Effect of reproductive status on the probability of bone failure. Survival curves for femurs from nonbreeders versus queens (Wald test, p=0.02, n = 34) and versus queens with ≥6 offspring (Wald test, p=0.006, n = 28), based on predictions from the midshaft cortical area and data from *Jepsen et al., 2003*. Vertical dashed lines indicate group medians, with the median failure time for nonbreeders fixed at a value of 1.

The online version of this article includes the following figure supplement(s) for figure 5:

**Figure supplement 1.** Relationship between max load and cortical area in mouse femurs.

In addition to previously described vertebral growth, we found that queen Damaraland mole-rats lose bone lengthening potential in the long bones and develop thinner femurs that are predicted to be more prone to mechanical failure. Moreover, gene expression levels in queens reflect a signature of bone resorption, rather than bone growth, at the time of sampling, which occurred 1–2 years post-transition. The molecular signature of bone resorption temporally aligns with changes in morphology, in which accelerated vertebral growth primarily occurs during a female's first pregnancy, whereas cortical thinning in the long bones is a function of repeated cycles of offspring production. Thus, queens quickly progress from traits typically associated with pre-reproductive and pubertal growth in mammals (e.g., body elongation), to traits typically linked to aging (e.g., marrow cavity expansion and cortical thinning).

The complex pattern of bone growth and bone resorption in queens likely involves multiple regulatory mechanisms. Because estrogen is known to impact bone growth and maintenance (*Cutler, 1997*; *Khalid and Krum, 2016*), and estrogen levels are higher in mole-rat queens relative to nonbreeding females (*Bennett, 1994*), we hypothesized that queens and nonbreeders would differ in their response to estradiol in bone marrow-derived cells. Surprisingly, we observed no gene expression response to estradiol treatment. By itself, this result could be a function of the specific concentration or duration of estradiol treatment we applied. However, we also observed no enrichment for estrogen receptor binding motifs near queen upregulated genes, and no evidence that estrogen or androgen receptor binding sites are differentially bound in cells from queens versus nonbreeders. Thus, our results suggest a role for other, as-yet unknown signaling pathways in the queen-specific signature of long bone cortical resorption (although it does not exclude the possibility that estrogen signaling influences other phenotypes, such as bone elongation, growth plate closure, or collagen organization, which require further study; *Juul, 2001*; *Cake et al., 2005*; *Ham et al., 2002*).

Bone loss in Damaraland mole-rat queens may be an extreme of the typical mammalian pattern of bone remodeling, in which bone mineral density decreases during pregnancy and lactation, but recovers once offspring are weaned (*Kovacs, 2016*). Thinning in mole-rats may be sustained, however, because queens can begin gestating soon after lactating for the previous litter, leaving little to no time for recovery. One possible reason that this fast rate of breeding is achievable is that queens in colonies with more helpers work less and rest more (*Houslay et al., 2020*), consistent with studies in other cooperative species that show that helpers alleviate breeding-associated efforts (*Bales et al., 2000*; *Clutton-Brock and Manser, 2016*; *Crick, 1992*; *Paquet et al., 2013*; *Russell, 2003*; *Scantlebury et al., 2002*). Paradoxically, helpers might not only help offset costs of, but also contribute to, decreased bone mass in queens, given that large numbers of helpers are themselves produced via high queen fecundity, and reduced physical activity can also lead to decreases in bone mass (*Morseth et al., 2011*).

The extent to which helpers reduce the costs of breeding to queens may also differ between species depending on the relative numbers of helpers to breeders. For example, in eusocial insects, large colonies and the high ratio of helpers to queens reduce the costs of reproduction to queens to very low levels (*Wilson, 1971*; *Keller and Genoud, 1997*). Similarly, in naked mole-rats (where colonies can include hundreds of animals compared to dozens in Damaraland mole-rat colonies; *Jarvis, 1981*; *Jarvis and Bennett, 1993*; *Jarvis et al., 1994*), a small sample of queens (n = 6) suggests increased rather than decreased femoral cortical thickness relative to age-matched nonbreeders (*Pinto et al., 2010*). Testing how the costs and benefits of reproduction are resolved across different levels of cooperativity, including the molecular mechanisms that mediate these differences, is an important next step towards understanding the evolution of cooperative breeding in mammals.

Finally, despite frequent analogies between Damaraland mole-rats and eusocial insects (*Jarvis and Bennett, 1993*; *Jarvis et al., 1994*; *Burda et al., 2000*), our results suggest some key points of differences. Specifically, while abdominal lengthening allows queen mole-rats to increase fecundity per reproductive effort, loss of cortical bone in the femur is unlikely to directly benefit either fertility or survival. Instead, it reflects the cumulative burden of continuous cycles of pregnancy and lactation (*Kovacs, 2016*). Thus, unlike eusocial insect queens (*Rodrigues and Flatt, 2016*; *Rueppell et al., 2016*), Damaraland mole-rat queens incur morphological costs to concentrated reproduction in addition to morphological changes that facilitate increased fitness. How these costs translate into fertility or survival outcomes in natural populations remains a fascinating, unanswered question.

# Materials and methods

## Key resources table

| Reagent type (species) or resource | Designation | Source or reference | Identifiers | Additional information |
|---|---|---|---|---|
| Peptide, recombinant protein | Recombinant human fibroblast growth factor-basic | Biocom Africa Biotech (http://www.biocombiotech.com) | Cat # 571504 | |
| Chemical compound, drug | Y-27632 Rock Inhibitor | Cayman Chemical | Cat # 10005583 | |
| Software, algorithm | R Project for Statistical Computing | R Project for Statistical Computing | RRID:SCR_001905 | |
| Software, algorithm | cutadapt | cutadapt | RRID:SCR_011841 | |
| Software, algorithm | HTSeq | HTSeq | RRID:SCR_005514 | |
| Software, algorithm | STAR | STAR | RRID:SCR_004463 | |
| Software, algorithm | LIMMA | LIMMA | RRID:SCR_010943 | |
| Software, algorithm | DESeq | DESeq | RRID:SCR_000154 | |
| Software, algorithm | G:Profiler | G:Profiler | RRID:SCR_006809 | |
| Software, algorithm | GATK | GATK | RRID:SCR_001876 | |
| Software, algorithm | VCFtools | VCFtools | RRID:SCR_001235 | |
| Software, algorithm | BEAGLE | BEAGLE | RRID:SCR_001789 | |
| Software, algorithm | CIBERSORT | CIBERSORT | RRID:SCR_016955 | |
| Software, algorithm | Trim Galore | Trim Galore | RRID:SCR_011847 | |
| Software, algorithm | BWA | BWA | RRID:SCR_010910 | |
| Software, algorithm | MACS | MACS | RRID:SCR_013291 | |
| Software, algorithm | HOMER | HOMER | RRID:SCR_010881 | |
| Software, algorithm | MATLAB | MATLAB | RRID:SCR_001622 | |
| Software, algorithm | ImageJ | ImageJ | RRID:SCR_003070 | |
| Software, algorithm | Avizo 3D Software | Avizo 3D Software | RRID:SCR_014431 | |
| Other | MEM-alpha | Sigma-Aldrich | Cat # M4526 | |
| Other | Hyclone Research Grade Fetal Bovine Serum (FBS), South American (Colombia) origin, IRRADIATED | Separations (South Africa; http://separations.co.za) | SV30160.03IR | |

## Study system and experimental design

Damaraland mole-rats (*F. damarensis*) were maintained in a captive colony at the Kuruman River Reserve in the Northern Cape Province of South Africa, within the species' natural range. With the exception of the predictive Euler characteristic transform analysis (which included two nonbreeding females born in the wild and subsequently maintained in captivity), only animals born in captivity, with known birthdates, ages, and litter composition, were used in this study. Animals were maintained in artificial tunnel systems built from PVC pipes with compartments for a nest-box and waste-box and transparent windows to allow behavioral observation (*Zöttl et al., 2016*). Animals were fed ad libitum with sweet potatoes and cucumbers.

Adult females (>1 year) from 16 natal colonies were randomly assigned to be either nonbreeders or queens, such that females assigned to queen status had age-matched littermates, where possible, who were assigned to the nonbreeding condition. To distinguish the effects of queen status from release from reproductive suppression, nonbreeders were either maintained in their natal colony as helpers or maintained alone, which models the social condition experienced by dispersing females. Females assigned to the breeder condition were transferred to a new tunnel system with an unrelated male from a separate social group. Nine new breeding females, six helpers, and eight solitaire females (age-matched littermates where possible; *Supplementary file 1*) were set up in December 2015–July 2016 (*Thorley et al., 2018*). With one exception (animal G10F026), animals maintained their breeding status for 14–22 months before sample collection. One queen and five helpers that were siblings, but not age-matched littermates, of experimental animals were also included in sample collection. To increase the final sample size, an additional four breeding colonies, matched against four age-matched littermate helpers, were formed in October 2017 and followed for 11–12 months (*Supplementary file 1*). One queen died before sample collection, and one nonexperimental helper was euthanized during the course of the study and included in sample collection. The final sample size included 13 queens, 15 helpers, and 8 solitaire females.

## X-ray data

For a subset of study subjects, full body X-rays were taken using the Gierth TR 90/20 battery-operated generator unit with portable Leonardo DR Mini plate (OR Technology, Rostock, Germany) every 2 months during the first 12 months of the experiment and at the time of sacrifice. From these X-rays, an experimenter blind to animal breeding status measured the length of each LV (from vertebra 1 to 7), the right femur, the right tibia, body length, and the width of the zygomatic arch using ImageJ (*Schneider et al., 2012*). The caudal-most LV was labeled as LV7. We tested for an effect of breeding status on LV5 using a linear mixed model in which post-pairing time point, breeding status, and the interaction of time point by breeding status were modeled as fixed effects and animal ID as a random effect.

## Effects of queen body length on fertility

To test the effect of maternal body length on litter size and pup size, we used body length measurements obtained during routine colony monitoring of all queens maintained in the captive colony (i. e., not restricted to experimental animals). Following *Thorley et al., 2018*, we used body length measurements obtained nearest to, and no more than 90 days from, the date of parturition. The resulting data set included 328 litters (971 pups) from 76 mothers, which represents a 76% increase over an earlier analysis of this relationship in *Thorley et al., 2018*. We fit two linear mixed effects models. In the first model, we modeled litter size as a function of maternal body length, controlling for whether the litter was the female's first litter, and included maternal ID as a random effect. In the second model, we modeled pup mass as a function of maternal body length, controlling for litter size and whether the litter was the female's first litter as fixed effects, and maternal ID and litter ID as random effects.

## Sample collection and cell culture from LV and long bones

Animals were deeply anesthetized with isoflurane and sacrificed with decapitation following USGS National Wildlife Health Center guidelines and under approval from the Animal Ethics Committee of the University of Pretoria. Immediately upon sacrifice, the LV and long bones were dissected, and attached muscle tissue was removed with forceps. LV6 and 7 and the right femur and tibia were

collected into 50% ethanol for 24 hr, then transferred to 70% ethanol and stored at 4° C for μCT scans and histochemistry.

To isolate bone cells for culture, LV1–5 were incubated in 2% Collagenase P (Roche, Switzerland) for 30 min at 30° C. Each bone was then cut in half and transferred to a 1.5 ml microcentrifuge tube containing a G-Tube microcentrifuge tube (VWR, Radnor, PA, USA) that had been punctured at the bottom with a 15 gauge needle. Tubes were spun at 3000 RCF for 5 s, allowing the marrow to collect into the 1.5 ml microcentrifuge tube. Cell pellets were resuspended in red blood cell lysis buffer, pooled, and incubated for 3 min at room temperature. 10 ml bMSC medium (MEM-alpha [Sigma-Aldrich, St. Louis, MO, USA] + 15% fetal bovine serum [Hyclone, Logan, UT, USA] + 1% penicillin/streptomycin + 2 ng/ml recombinant human fibroblast growth factor-basic [Biocam, Centurion, Gauteng, South Africa] + 10 nM ROCK inhibitor Y-27632 [RI; Cayman Chemical, Ann Arbor, MI, USA]) was added to stop lysis, and the tubes were spun for 5 min at 300 RCF. The cell pellet was resuspended in 1 ml bMSC medium and strained through a 70 μm cell strainer. Cells were plated at $1.6 \times 10^5$ cells per $cm^2$. The long bones (excluding right femur and tibia) were processed to enrich for bMSCs following the same procedure, but without incubation in Collagenase P. Cells were cultured at 37°C and 5% $CO_2$. 24 hr post plating, plates were carefully washed three times with $1\times$ PBS and supplied with fresh medium to remove nonadherent cells. Once bMSC clusters were visible (2–9 days post plating), plates were fed bMSC medium without RI or fed bMSC medium without RI +10 nM estradiol (E2). 24 hr later, cells were collected into buffer RLT and stored at −80°C. Samples were shipped on dry ice to Duke University for RNA extraction using the Qiagen RNeasy Micro Kit. RNA-Seq libraries were generated using the NEBNext Single Cell/Low Input RNA Library Prep Kit for Illumina.

## Gene expression analysis

RNA-Seq libraries were sequenced on an Illumina HiSeq 4000 (100 basepair single-end reads) to a mean coverage of 16.1 ± 3.9 s.d. million reads. Reads were trimmed with *cutadapt* version 2.3 (*Martin, 2011*) (RRID:SCR_011841; parameters: -q 20 -e 0.2 `--times` 5 `--overlap` 2 -a AGATCGGAA-GAGC -a 'T' `--minimum-length`=20). Trimmed reads were then mapped to the Damaraland mole-rat v1.0 genome (*Fang et al., 2014*) (DMR_v1.0) using two pass mapping with STAR (RRID:SCR_004463) (*Dobin et al., 2013*). Only uniquely mapped reads were retained. HTseq (RRID:SCR_005514) (*Anders et al., 2015*) was used to quantify read counts mapping to genes (using the v1.0.92 gtf file from Ensembl; we extended the genomic coordinates of the *SERPINE1* gene by 2000 basepairs in both directions due to very high expression directly adjacent to the annotated coordinates). We transformed read counts to transcripts per million (TPM) (*Wagner et al., 2012*) and retained only genes with TPM ≥2 in at least 25% of samples. We performed voom normalization (RRID:SCR_010943) (*Law et al., 2014*) on the raw counts using normalization factors produced by the trimmed mean of M-values (TMM) method (*Robinson and Oshlack, 2010*) in DESeq (RRID:SCR_000154) (*Anders and Huber, 2010*). We used the *limma* (*Smyth, 2005*) function *lmFit* to regress out the proportion of uniquely mapped reads in genes (which controls for efficiency of mRNA selection during RNA-Seq library preparation) and animal natal colony (which controls for littermate sets and date of sacrifice) to obtain normalized, batch-corrected gene expression values for downstream analysis. We used the mixed effects model approach in *emmreml* (*Akdemir and Okeke, 2015*) to estimate, for each gene, the effect of breeding status on gene expression within LV and within long bones using the following model:

$$y_1 = \mu + d_i\beta_1 + b_i\beta_2 + q_i\beta_3 * I(b=0) + q_i\beta_4 * I(b=1) + s_i\beta_5 * I(b=0) + s\beta_6 * I(b=1) + Zu + \varepsilon_i,$$

$$u \sim MVN\, 0, \sigma_u^2 K$$

$$\varepsilon \sim MVN\, 0, \sigma_e^2$$

where *y* is the vector of gene expression levels for *n* = 47 samples (indexed by *i*); μ is the intercept; *d* is the number of days in culture and $\beta_1$ its effect size; *b* is bone type (i.e., long bones or LV) and $\beta_2$ its effect size; and *q* is a 0/1 variable representing breeder status and $\beta_3$ and $\beta_4$ its effect size in long bones and LV, respectively. *I* is an indicator variable for bone type (0 = long bone; 1 = LV). *s* is a 0/1 variable representing whether the cells were cultured with estradiol and $\beta_5$ and $\beta_6$ are its effect sizes

in long bones and LV, respectively. Z is an incidence matrix that maps samples to animal ID to take into account repeated sampling from the same individual, and $u$ is a random effect term that controls for relatedness. K is an $m$ by $m$ matrix of pairwise relatedness estimates (derived from the genotype data, described below) between all $m$ animals. $\varepsilon$ is the residual error, $\sigma_u^2$ is the genetic variance component, and $\sigma_e^2$ is the environmental variance component. We also ran an identical model but with an additional fixed effect of solitaire status in long bones and in LV, to test for a difference in gene expression between helpers and solitaires. To control for multiple testing, we calculated the FDR following *Storey and Tibshirani, 2003* using an empirical null distribution derived from 100 permutations of each variable of interest.

We used g:profiler (RRID:SCR_006809) (*Raudvere et al., 2019*) to perform Gene Ontology enrichment analysis of the genes upregulated with queen status in LV and long bones (151 of 171 genes significantly associated with queen status at a 10% FDR). All genes in the original analysis set were used as the background gene set. We set both the minimum size of the functional category and the minimum size of the query/term intersection to 3. Finally, we retained categories that passed a Bonferroni-corrected p-value of 0.05.

## Genotyping to estimate relatedness

To control for relatedness when modeling the gene expression data, we performed single-nucleotide polymorphism (SNP) genotyping of the RNA-Seq data using the Genome Analysis Toolkit (*McKenna et al., 2010*) (GATK; RRID:SCR_001876). We used the SplitNCigarReads function on the trimmed, uniquely mapped reads and performed GATK indel realignment. Base recalibration was performed by using all SNPs with GQ $\geq$4 in an initial UnifiedGenotyper run on the full data set as a reference. Genotypes were called on the recalibrated bam files using HaplotypeCaller. Variants were filtered with the following GATK VariantFiltration parameters: QUAL < 100.0, QD < 2.0, MQ < 35, FS > 30, HaplotypeScore > 13, MQRankSum < −12.5, ReadPosRankSum < −8. Variants were further filtered with *vcftools* (RRID:SCR_001235) (*Danecek et al., 2011*) to only retain biallelic SNPs in Hardy–Weinberg equilibrium (p>0.05) with minor allele frequency $\geq$ 0.1, minimum mean depth of 5, max missing count of 2, and minimum GQ of 99. Finally, SNPs were thinned to a distance of 10 kb basepairs, resulting in a final data set of 1965 stringently filtered biallelic SNPs. Missing values were imputed using *beagle* (RRID:SCR_001789) (*Browning and Browning, 2007*), and the resulting vcf file was used to create a kinship matrix using *vcftools* (*Danecek et al., 2011*). Values of the kinship matrix were confirmed to be higher in known siblings compared to non-siblings (unpaired t-test, t = 27.939, p=2.23×10$^{-12}$; means = 0.513 and −0.097). Two pairs of siblings were found to have different fathers (G1F022 and G1F025; G4F020 and G4F019).

## Cell-type heterogeneity

Although selection for adherent cells from bone marrow enriches for bMSCs, other cell types are also present (*Phinney et al., 1999*). To assess whether cell-type heterogeneity accounts for queen-associated differential expression, we used CIBERSORT (RRID:SCR_016955) to deconvolve the proportion of component cell types from the RNA-Seq data (*Newman et al., 2015*). We trained CIBERSORT on a data set of quantile normalized gene expression values from mouse purified primary cell populations (*Hume et al., 2010*). Specifically, we subset the training data to 27 purified cell populations of mesenchymal or hematopoietic origin (*Figure 2—figure supplement 2*) and to genes that were included in our mole-rat gene expression data set. We then predicted the composition of the cells that contributed to the mole-rat quantile normalized gene expression data set, for each sample.

To test whether cell-type heterogeneity was significantly explained by queen status, we also modeled cell-type proportion (as summarized by the first principal component of CIBERSORT-estimated proportions for all 27 potential cell types; PC1 explains 50.9% of the overall variation) following the same method used for gene-by-gene expression analysis but with PC1 included as an explanatory variable. We then performed mediation analysis on each of the 171 genes that showed a significant effect of breeding status at FDR < 10%. To do so, we first estimated the indirect effect of breeding status on gene expression through the mediating variable (CIBERSORT PC1). The indirect effect of breeding status through CIBERSORT PC1 was estimated by calculating the difference in the effect of breeding status between two models: one model that did not include the mediator (i.e., $\beta_3$ and $\beta_4$

from *Equation 1*) and the same model with the addition of the mediating variable. We performed 1000 iterations of bootstrap resampling to obtain the 95% confidence interval for the indirect effect and considered an indirect effect for a gene significant if the 95% interval did not overlap 0.

## ATAC-seq data and transcription factor binding site analysis

To investigate whether differentially expressed genes were associated with accessible binding motifs for specific transcription factors, we generated Omni-ATAC-seq data to profile regions of open chromatin (*Buenrostro et al., 2013*; *Corces et al., 2017*). We performed Omni-ATAC-seq on both LV bMSCs and long bone bMSCs from two female nonbreeding and two queen mole-rats (n = 8 libraries total), following the published protocol (*Corces et al., 2017*) with the following modifications: 5000 cells were centrifuged at 500 RCF for 5 min at 4°C. The pellet was resuspended in 50 µl transposition mix (25 µl 2× TD buffer, 16.5 µl PBS, 6.75 µl water, 1 µl 10% NP40, 1 µl 10% Tween-20, 1 µl 1% digitonin, and 0.25 µl Tn5 transposase). The reaction was incubated at 37°C for 30 min without mixing, followed by a 1.5× Ampure bead cleanup. Omni-ATAC libraries were sequenced on a NovaSeq 6000 as 100 basepair paired-end reads to a mean coverage (± SD) of 26.9 (± 4.4) million reads (range: 16.8–38.3). Reads were trimmed with Trim Galore! (RRID:SCR_011847) (*Krueger, 2015*) to remove adapter sequence and low-quality basepairs (Phred score <20; reads ≥25 bp). Read pairs were mapped to the DMR_v1.0 genome using *bwa-mem* (RRID:SCR_010910) (*Li and Durbin, 2010*) with default settings. Only uniquely mapped reads were retained. The alignment bam files for each treatment (breeding or nonbreeding) were merged, and open chromatin regions were identified using MACS2 v2.1.2 (RRID:SCR_013291) (*Zhang et al., 2008*) with the following parameters: '-nomodel -keep-dup all -q 0.05'. We combined open chromatin peaks with regions in the DMRv1.0 genome that match sequences of vertebrate transcription factor binding site motifs, using motifs defined in the HOMER database (RRID:SCR_010881) (*Heinz et al., 2010*). We used Fisher's exact tests (using a p-value threshold of 0.01) to test if TFBMs belonging to the same transcription factor were enriched in open chromatin regions within 2000 bp of the 5′ most transcription start site of queen upregulated genes.

To compare genome-wide signatures of DNA-transcription factor binding for AR, ESR1, and ESR2, we characterized transcription factor footprints in queens and nonbreeders, in both the LV and long bones, using HINT-ATAC from the Regulatory Genomics Toolbox (RGT) with default parameters (*Li et al., 2019*). We focused on the subset of peak regions called using MACS2 (*Zhang et al., 2008*). We identified TF footprints by merging reads within each bone type-breeding status combination and calling footprints on the combined data. For each bone type, we then created a meta-footprint set by merging the respective footprint calls across queens and nonbreeders using the *bedtools* function *merge* (*Quinlan and Hall, 2010*). We identified transcription factor motifs in the *DMR_v1.0* genome that fell within meta-footprints, based on the JASPAR CORE Vertebrates set of curated position frequency matrices (*Sandelin et al., 2004*). Finally, we tested for differential footprints of AR, ESR1, and ESR2 binding using the RGT *differential* function, using the activity score metric described in *Li et al., 2019* and default parameters.

## Micro-CT scans and analysis

We performed µCT scans of LV6, LV7, right femur, and right tibia using a VivaCT 80 scanner (Scanco Medical AG, Brüttisellen, Switzerland) set at 55 kVp and 145 µA, with voxel size 10.4 µm. Trabecular bone was quantified using direct values (i.e., 'No model') from the 100 µCT slices below the proximal tibia growth plate, the 100 µCT slices above the distal femur growth plate, and the 100 µCT slices medial to the caudal growth plate of LV6. To obtain midshaft cross-sections of the femur, tibia, and LV6, we first reduced each bone mesh to 100,000 faces using Avizo Lite version 9.7.0. Mesh files from the same bone type were auto-aligned using Auto3dgm (*Boyer et al., 2015*) in MATLAB (RRID:SCR_001622). Aligned mesh files were then back scaled to their original sizes in MATLAB, and the midshaft cross-section was generated using Rhinoceros version 6. LV6 cross-sections required manual segmentation, which was performed in Adobe Illustrator CC version 23.0.2. The MomentMacro plugin in ImageJ (RRID:SCR_003070) was used to calculate bone area, minimum second moment of inertia, and polar second moment of area.

## Classification of breeding status from bone shape

To predict breeding status from bone shape, we reduced each bone mesh to 100,000 faces using Avizo Lite version 9.7.0 (RRID:SCR_014431). Mesh files from the same bone type were auto-aligned using Auto3dgm (*Boyer et al., 2015*) in MATLAB. The resulting aligned and scaled mesh files were used as input to perform the unsmooth Euler characteristic transform (ECT) for LV6 and to perform the smooth Euler characteristic transform (SECT) for femurs (*Crawford et al., 2016*) (as performance was optimized for femurs by including smoothing).

The ECT represents each shape as a two-parameter function of direction and height. We evaluate these functions at a discrete lattice of points, so that each point is represented as a vector. For each bone type, we calculated the Euler characteristic curve in 162 directions distributed approximately uniformly across the shape. For each of the 162 Euler characteristic curves, we ran 100 height parameter evaluations for LV6 and an average of 71 height parameter evaluations for the femur, so that each LV6 (femur) was represented with a vector of length 16,200 (11,423). Using the ECT data, we performed leave-one-out predictions, running each bone type separately, using the linear kernel and c-classification with the support vector machine (SVM) implemented by the R package *e1071* (*Dimitriadou et al., 2008*). The SVM classifier was equipped with 1:100,000 weighting to balance for the different number of breeders and nonbreeders in the sample. The empirical p-values were estimated for each bone type by running 100 permutations of the queen/nonbreeder labels (*Golland et al., 2005*).

We note that the ECT does not provide information about specific regions of a shape, but rather is a topological summary statistic that captures the geometric complexity of a shape by quantifying the shape's number of connected components, voids, and closed loops. This approach avoids the need for landmarks, which may be missing or vulnerable to observer error in some data sets (*Crawford et al., 2016*). The ECT is amenable to regression models and is therefore useful for testing whether shape significantly predicts the value of an outcome variable (here, breeding status). However, because it does not identify the regions of the shape that contribute most to predictive power, the regions of the LV and femur that most strongly differentiate queens from nonbreeders remain open to further study.

## Histochemistry

For a subset of individuals (*Supplementary file 9*), the tibia and LV7 were plasticized, sectioned, and stained with Safranin O by the Washington University Musculoskeletal Research Center. The proportion of the tibia proximal growth plate that was fused, and the mean proportion of the LV7 cranial and caudal growth plates that were fused, were measured in ImageJ from Safranin O-stained sections. To quantify growth plate activity, we calculated the number of chondrocyte columns (defined as linear stacks of at least three chondrocytes) controlling for length of open growth plate. For each bone type (tibia and LV7), we ran two models: proportion of growth plate fusion or chondrocyte columns per mm growth plate as the dependent variable, and number of offspring born and age as the independent variables.

## Cortical thickness across the femur

We used Stradview (*Treece, 2019*; *Treece et al., 2010*) on dicom images from the μCT scans to measure and visualize, in an automated manner, cortical thickness across the surface of the femur. Bone surface was defined in Stradview by thresholding pixel intensity and contouring the bone at every 14 sections, with the following parameters: resolution = medium, smoothing = standard, strength = very low, contour accuracy = 6. To measure cortical thickness, we used the auto threshold method in Stradview, with line width set to 5, smooth set to 1, and line length set to 3 mm. The smoothed thickness values of each femur were then registered (i.e., mapped) to a single 'canonical' femur surface (mole-rat GRF002) using wxRegSurf v18 (http://mi.eng.cam.ac.uk/~ahg/wxRegSurf/). We sectioned the cortical thickness values into deciles according to location along the length of the femur. The top and bottom deciles were removed because cortical and trabecular bone towards the ends of the femur could not be easily differentiated by the automated method. Deciles were then recreated for the remaining length of the bone (i.e., the central 80%). From each bone decile, we estimated cortical thickness as the mean of all cortical thickness measures within that interval. For

each decile across animals, we used a linear mixed model to model cortical thickness as a function of breeding status and number of offspring, with litter pair as a random effect.

## Modeling the probability of bone failure

Previous research on mechanical properties of rodent femurs found that, among several morphological and compositional traits measured in eight morphologically varying mouse strains, CA at the midshaft was the best predictor of maximum load (defined as the greatest force attained prior to bone failure, measured via four-point bending; published Pearson's $r$ = 0.95) (*Jepsen et al., 2003*). We therefore used CA at the femoral midshaft to predict max load of Damaraland mole-rat femurs. To do so, we first fit a linear model of max load as a function of CA (unadjusted for body weight) using published mouse data ($R^2$ = 0.877, n = 81, p=$6.64\times10^{-38}$) (*Jepsen et al., 2003*). We extrapolated from this linear fit to predict max load from CA at the midshaft of Damaraland mole-rat femurs. Predicted max loads were then used as input for Cox proportional hazards models, comparing either all queens to nonbreeders or queens with $\geq$6 offspring to nonbreeders. Models were fit using the R function *coxph* and were confirmed to meet the proportional hazards assumption using the *cox.zph* function in the R package *survival* (*Therneau, 2020*). Because max load was not directly measured in the Damaraland mole-rats, we used the Cox proportional hazards models to specifically evaluate the relative hazard of bone failure depending on queen status/number of offspring. We therefore report the results in *Figure 5* based on relative force (with the median predicted failure value for nonbreeders set to 1) instead of absolute force in Newtons. We note that the analysis makes an important assumption that the linear relationship between CA and maximum load observed in mice is shared with Damaraland mole-rats. Future mechanical loading tests of mole-rat bones, which were not possible in this study, will therefore be important for validating and refining the present modeling results.

## Acknowledgements

We thank Tim Vink, Dave Gaynor, and the mole-rat house staff and volunteers for their tremendous contributions to the Kalahari Mole-rat Project. We also thank Irene Garcia, Mari Cobb, Brianna Bowman, Anna Luiza Wolf, Alice Zhou, Yilin Yu, BJ Nielsen, Tawni Voyles, and Lorin Crawford for their contributions to sample collection, data generation, and modeling, Graham Treece for guidance on quantifying bone thickness with Stradview, Karl Jepsen for sharing data on mouse femurs, Lou DeFrate for guidance on estimating bone strength, Saideep Gona and Luis Barreiro for their contributions on the footprint analysis, and members of the Tung lab for feedback on earlier versions of this manuscript. Support for this work was provided by the Human Frontier Science Program (RGP0051-2017 to JT, SM, and TCB), the National Science Foundation (IOS-7801004 to JT), the National Institutes of Health (AR076325 and AR071967 to CK, F32HD095616 to RAJ), a Sloan Foundation Early Career Research Fellowship to JT, a Foerster-Bernstein Postdoctoral Fellowship to RAJ, and a Natural Environmental Research Council Doctoral Training Program to JaT. This research is part of a project that has received funding from the European Research Council (ERC) under the European Union's Horizon 2020 research and innovation programme (Grant agreement No. 294494 and 742808 to TCB). High-performance computing resources were supported by the North Carolina Biotechnology Center (Grant Number 2016-IDG-1013).

## Additional information

### Competing interests

Jenny Tung: Jenny Tung is a Reviewing Editor at eLife. The other authors declare that no competing interests exist.

### Funding

| Funder | Grant reference number | Author |
|---|---|---|
| European Research Council | 294494 | Tim Clutton-Brock |

| European Research Council | 742808 | Tim Clutton-Brock |
|---|---|---|
| Human Frontier Science Program | RGP0051-2017 | Sayan Mukherjee<br>Tim Clutton-Brock<br>Jenny Tung |
| National Science Foundation | IOS-7801004 | Jenny Tung |
| National Institutes of Health | F32HD095616 | Rachel A Johnston |
| Alfred P. Sloan Foundation | | Jenny Tung |
| Foerster-Bernstein Postdoctoral Fellowship | | Rachel A Johnston |
| Natural Environment Research Council | Doctoral Training Program Grant | Jack Thorley |
| North Carolina Biotechnology Center | 2016-IDG-1013 | Jenny Tung |
| National Institutes of Health | AR076325 | Courtney M Karner |
| National Institutes of Health | AR071967 | Courtney M Karner |

The funders had no role in study design, data collection and interpretation, or the decision to submit the work for publication.

## Author contributions
Rachel A Johnston, Conceptualization, Data curation, Formal analysis, Investigation, Visualization, Methodology, Writing - original draft, Writing - review and editing; Philippe Vullioud, Jack Thorley, Leyao Shen, Investigation, Writing - review and editing; Henry Kirveslahti, Sayan Mukherjee, Formal analysis, Investigation, Writing - review and editing; Courtney M Karner, Resources, Investigation, Methodology, Writing - review and editing; Tim Clutton-Brock, Conceptualization, Supervision, Funding acquisition, Investigation, Writing - review and editing; Jenny Tung, Conceptualization, Resources, Supervision, Funding acquisition, Investigation, Writing - original draft, Writing - review and editing

## Author ORCIDs
Rachel A Johnston https://orcid.org/0000-0002-8965-1162
Jenny Tung https://orcid.org/0000-0003-0416-2958

## Ethics
Animal experimentation: Animals were deeply anesthetized with isoflurane and sacrificed with decapitation following USGS National Wildlife Health Center guidelines and under approval from the Animal Ethics Committee of the University of Pretoria (Permit #EC081-17).

## Decision letter and Author response
Decision letter https://doi.org/10.7554/eLife.65760.sa1
Author response https://doi.org/10.7554/eLife.65760.sa2

# Additional files

## Supplementary files
• Supplementary file 1. Table summarizing study animals.

• Supplementary file 2. Table of results of mixed effects model of mole-rat gene expression data testing for effect of solitaire versus helper social status. bone_0 refers to long bones; bone_1 refers to lumbar vertebrae.

• Supplementary file 3. Table of results of multivariate model explaining litter size (first model) or pup mass (second model).

• Supplementary file 4. Table of results of mixed effects model of mole-rat gene expression data. bone0 refers to long bones; bone1 refers to lumbar vertebrae.

• Supplementary file 5. Table of proportions of cell types estimated with CIBERSORT, based on reference gene expression levels for 412 marker genes in 27 purified mouse cell types (*Hume et al., 2010*).

• Supplementary file 6. Table of 95% confidence intervals of mediation analysis testing for cell-type proportions as mediating the effect of queen status on gene expression (in long bones or in lumbar vertebrae).

• Supplementary file 7. Table of Gene Ontology (GO) enrichment results of genes upregulated with queen status.

• Supplementary file 8. Table of transcription factor binding motifs enriched in open chromatin regions near genes upregulated with queen status.

• Supplementary file 9. Table of sample info for bone sections stained with Safranin O.

• Supplementary file 10. Table of effects of number of total offspring on mean cortical thickness per femur shaft decile.

• Transparent reporting form

### Data availability

All RNA sequencing data generated during this study are available in the NCBI Gene Expression Omnibus (series accession GSE152659). ATAC-Seq data are available in the NCBI Sequence Read Archive (BioProject accession number PRJNA649596). µCT data from this study are available on MorphoSource (http://www.morphosource.org, project 1056). All code used for the study are available at copy archived at [https://archive.softwareheritage.org/swh:1:rev:f10ef475fd79fc69aa44496ce8ffac5dc7f5abf9].

The following datasets were generated:

| Author(s) | Year | Dataset title | Dataset URL | Database and Identifier |
|---|---|---|---|---|
| Johnston RA, Vullioud P, Thorley J, Kirveslahti H, Shen L, Mukherjee S, Karner C, Clutton-Brock T, Tung J | 2021 | Morphological and genomic shifts in mole-rat queens increase fecundity but reduce skeletal integrity | https://www.ncbi.nlm.nih.gov/geo/query/acc.cgi?acc=GSE152659 | NCBI Gene Expression Omnibus, GSE152659 |
| Johnston RA, Vullioud P, Thorley J, Kirveslahti H, Shen L, Mukherjee S, Karner C, Clutton-Brock T, Tung J | 2021 | Morphological and genomic shifts in mole-rat queens increase fecundity but reduce skeletal integrity | https://www.ncbi.nlm.nih.gov/sra/PRJNA649596 | NCBI Sequence Read Archive, PRJNA649596 |
| Johnston RA, Vullioud P, Thorley J, Kirveslahti H, Shen L, Mukherjee S, Karner C, Clutton-Brock T, Tung J | 2020 | Assessment of effects of breeding status on bone morphology of female Damaraland mole-rats (Fukomys damarensis) | https://www.morphosource.org/Detail/ProjectDetail/Show/project_id/1056 | MorphoSource, project 1056, 1056 |

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
