## [Decision Letter]

**Acceptance summary:**

This manuscript takes a deep dive into the skeletal effects of burrowing and eusocial Damaraland mole rats. By exploring the genetic and skeletal consequences of breeding restricted to a single queen with multiple and closely-timed pregnancies and lactation, this study offers a compelling story that will bolster textbooks on skeletal biology, mammalian evolution, and ethology. The results show the molecular mechanisms driving adaptive plasticity within the unusually expanded lumbar spine and thin limb bones of queens are an adaptive consequence of breeding status.

**Decision letter after peer review:**

Thank you for submitting your article "Morphological and genomic shifts in mole-rat 'queens' increase fecundity but reduce skeletal integrity" for consideration by *eLife*. Your article has been reviewed by 2 peer reviewers, and the evaluation has been overseen by a Reviewing Editor and George Perry as the Senior Editor. The reviewers have opted to remain anonymous.

Essential Revisions:

1) From Reviewer #1: I recommend generating a new first figure with an image of the queens and non-breeders along with a full skeleton (x-ray is fine). Because this work is a novel case-study in mole rats that I think belongs in textbooks, a figure illustrating the study taxon with a bit of introduction and then highlighting of the skeleton will potentially broaden the audience. I could easily see this work as part of skeletal biology course.

2) Please also address all of the smaller editing and clarification changes requested by both reviewers.

*Reviewer #1 (Recommendations for the authors):*

Additional Figure: I recommend generating a new first figure with an image of the queens and non-breeders along with a full skeleton (x-ray is fine). Because this work is a novel case-study in mole rats that I think belongs in textbooks, a figure illustrating the study taxon with a bit of introduction and then highlighting of the skeleton will potentially broaden the audience. I could easily see this work as part of skeletal biology course.

Figure 1 b: Change the shape of mean dots for queens such that they are different than that of non-breeders

Line 623, I believe the VivaCT uses a known-density phantom for comparisons. Is density data available? This isn't a requirement but might offer another compelling measure to support a conclusion that extends beyond just endosteal resorption.

Does the concentration and integrity of collagens shift with breeding status?

This is lovely work. Thank you for publishing. Best of luck.

*Reviewer #2 (Recommendations for the authors):*

I cannot comment on the gene regulation analyses or suitability of the protocols associated with those, as this is not my area of expertise. Specifically, I cannot comment on methods section line 500 – 620.

I have several minor comments, by line:

Ln331: "a correlate of minimum resistance to bending" would be more accurate description of Imin – it's the minimum second moment of inertia

Ln418: Also perhaps the extent to which this represents a recoverable insult given longevity in queens for this species, see:

Schmidt, C. M., Jarvis, J. U. M., and Bennett, N. C. (2013). The long-lived queen: reproduction and longevity in female eusocial Damaraland mole-rats (Fukomys damarensis). African Zoology, 48(1), 193-196. doi:10.1080/15627020.2013.11407583

Ln631: Imin and CA can be extracted directly from the cross-section in Rhino using Areamoment without need for ImageJ. but here you re-thresholded a.tif or converted an extracted.eps and then used that in ImageJ, correct? Was this done to manual segment the endosteal margin?

Ln632: this macro extracts many other biomechanical properties (Imax,Imin, J, CA, TA, Theta, Ix, Iy), what was the rationale for selecting Imin? For example, J (Polar second moment of area) is often considered the most relevant indicator of a bones' mechanical performance. See, for example:

Ruff, C. B., Trinkaus, E., Walker, A., and Larsen, C. S. (1993). Postcranial robusticity in Homo. I: Temporal trends and mechanical interpretation. American journal of physical anthropology, 91(1), 21-53.

Imin is perhaps least likely to be strongly correlated with body mass. See also Davies and Shaw:

Davies, T. G., and Stock, J. T. (2014). The influence of relative body breadth on the diaphyseal morphology of the human lower limb. American Journal of Human Biology, 26(6), 822-835.

Ln636: More information needs to be provided about how bone shape was evaluated. I assume this was done using the scaled mesh files (post-alignment) or was this run using the full-size meshes? How is the method computing differences between meshes? Is mesh face number standardized/reduced? What are the output data that are used to assess similarity? I did not find any information (or a figure illustrating) the differences in shape between the queens and non-breeders for either element – what were the main differences in shape that held predictive power in the sample?

Ln673: This is an interesting idea and I appreciate the authors' consideration of this aspect since it attempts to estimate the consequence of bone resorption. However, it may be useful to add a caveat on the assumptions associated with this approach – mainly, I am not sure how valid it may be to extrapolate a linear model based on body mass of mice in comparison to that of a mole-rat. Figure S2a indicates around 150 g body mass for the specimens in the study, and certainly well below 35 g for a mouse. Also, the weaker relationship between body mass and CA for these mole rats (cited earlier in the manuscript) and the likelihood that the relationship between body mass and CA may follow a different trajectory for the two species may impact the validity of this estimate.

Ln691: Please provide all code associated with analyses performed in R and Matlab.

---

## [Author Response]

Essential Revisions:1) From Reviewer #1: I recommend generating a new first figure with an image of the queens and non-breeders along with a full skeleton (x-ray is fine). Because this work is a novel case-study in mole rats that I think belongs in textbooks, a figure illustrating the study taxon with a bit of introduction and then highlighting of the skeleton will potentially broaden the audience. I could easily see this work as part of skeletal biology course.

Thank you, we have made this addition as two new panels in Figure 1.

2) Please also address all of the smaller editing and clarification changes requested by both reviewers.

We have made the requested edits/changes (see below for details).

Reviewer #1 (Recommendations for the authors):Additional Figure: I recommend generating a new first figure with an image of the queens and non-breeders along with a full skeleton (x-ray is fine). Because this work is a novel case-study in mole rats that I think belongs in textbooks, a figure illustrating the study taxon with a bit of introduction and then highlighting of the skeleton will potentially broaden the audience. I could easily see this work as part of skeletal biology course.

Thank you for the positive feedback! Figure 1 now includes images of a queen and helper mole-rat and an x-ray of a pregnant queen.

Figure 1 b: Change the shape of mean dots for queens such that they are different than that of non-breeders

Done. In this panel (now Figure 1D), queens are circles and non-breeders are diamonds.

Line 623, I believe the VivaCT uses a known-density phantom for comparisons. Is density data available? This isn't a requirement but might offer another compelling measure to support a conclusion that extends beyond just endosteal resorption.

The microCT scanner is indeed phantom calibrated. We have now analyzed both apparent and material density, finding that apparent density, but not material density, decreases in queens. These results, which are concordant with the findings on reduced bone volume due to loss on the endosteal surface, are now included in the main text (lines 264-268).

Does the concentration and integrity of collagens shift with breeding status?

Unfortunately, we do not know whether breeding status affects collagen concentration or integrity, because we did not process the bones in a manner that would allow us to readily assess collagen content. However, given previous findings that breeding status alters estrogen levels, and that estrogen levels can influence properties of cartilage, breeding status-associated changes in collagen properties are possible. We have revised the text to note this possibility for future work (lines 367-369).

This is lovely work. Thank you for publishing. Best of luck.Reviewer #2 (Recommendations for the authors):I cannot comment on the gene regulation analyses or suitability of the protocols associated with those, as this is not my area of expertise. Specifically, I cannot comment on methods section line 500 – 620.I have several minor comments, by line:Ln331: "a correlate of minimum resistance to bending" would be more accurate description of Imin – it's the minimum second moment of inertia

We have revised the description of I_min_ (lines 307-308).

Ln418: Also perhaps the extent to which this represents a recoverable insult given longevity in queens for this species, see:Schmidt, C. M., Jarvis, J. U. M., and Bennett, N. C. (2013). The long-lived queen: reproduction and longevity in female eusocial Damaraland mole-rats (Fukomys damarensis). African Zoology, 48(1), 193-196. doi:10.1080/15627020.2013.11407583

We have revised the text to acknowledge that, given the long lifespan of queens, potential morphological changes in wild animals must be either recoverable or, minimally, manageable for an extended period of time (lines 297-299).

Ln631: Imin and CA can be extracted directly from the cross-section in Rhino using Areamoment without need for ImageJ. but here you re-thresholded a.tif or converted an extracted.eps and then used that in ImageJ, correct? Was this done to manual segment the endosteal margin?

Thank you for noting this; we had not been aware that Rhino can measure these variables directly. While the long bones did not require manual segmentation, the vertebra midsections did. We have added this information to the methods section (lines 621-622).

Ln632: this macro extracts many other biomechanical properties (Imax,Imin, J, CA, TA, Theta, Ix, Iy), what was the rationale for selecting Imin? For example, J (Polar second moment of area) is often considered the most relevant indicator of a bones' mechanical performance. See, for example:Ruff, C. B., Trinkaus, E., Walker, A., and Larsen, C. S. (1993). Postcranial robusticity in Homo. I: Temporal trends and mechanical interpretation. American journal of physical anthropology, 91(1), 21-53.Imin is perhaps least likely to be strongly correlated with body mass. See also Davies and Shaw:Davies, T. G., and Stock, J. T. (2014). The influence of relative body breadth on the diaphyseal morphology of the human lower limb. American Journal of Human Biology, 26(6), 822-835.

We selected CA for analysis because of its high correlation with maximum load in mice (Jepsen et al., *Mammalian Genome*, 2013). We selected I_min_ based on the logic that reduced bending strength of a bone in the weakest direction, if present, would reflect the most consequential cost to the animal (i.e., vulnerability to bone fracture). Given the relevance of J to bone mechanical performance, we have now also analyzed J and added the results to the main text. Specifically, we report that J, like I_min_, is correlated with body mass in nonbreeders, but not in queens. In queens, J is significantly predicted by the number of offspring produced (lines 306-315).

Ln636: More information needs to be provided about how bone shape was evaluated. I assume this was done using the scaled mesh files (post-alignment) or was this run using the full-size meshes? How is the method computing differences between meshes? Is mesh face number standardized/reduced? What are the output data that are used to assess similarity? I did not find any information (or a figure illustrating) the differences in shape between the queens and non-breeders for either element – what were the main differences in shape that held predictive power in the sample?

We have revised the text to provide detail on how bone shape was evaluated for predicting breeder status (lines 627-650). We used the Euler characteristic transform for summarizing bone shape because it is amenable to regression models and does not rely on assignment of landmarks (Crawford et al., arXiv, 2016). However, this method does not provide information on which characters of the shape are most predictive. We now highlight this limitation in the text (lines 651-659). We too were interested in identifying which regions of the bones are most predictive of breeder status. However, the method we recently developed for this specific purpose, SINATRA (Wang et al., *bioRxiv*, 2019) does not accommodate paired data (i.e., queen-nonbreeder littermate/sibling pairs), which we have found is essential for identifying breeding status-associated differences in our other analyses. Thus, for the current study, we have focused on breeder-associated differences in volume, thickness, and their associated biomechanical properties, leaving detailed shape analysis for future work.

Ln673: This is an interesting idea and I appreciate the authors' consideration of this aspect since it attempts to estimate the consequence of bone resorption. However, it may be useful to add a caveat on the assumptions associated with this approach – mainly, I am not sure how valid it may be to extrapolate a linear model based on body mass of mice in comparison to that of a mole-rat. Figure S2a indicates around 150 g body mass for the specimens in the study, and certainly well below 35 g for a mouse. Also, the weaker relationship between body mass and CA for these mole rats (cited earlier in the manuscript) and the likelihood that the relationship between body mass and CA may follow a different trajectory for the two species may impact the validity of this estimate.

We agree that potential differences between species is an important caveat to consider, and now discuss this point in the text (lines 707-714).

Ln691: Please provide all code associated with analyses performed in R and Matlab.

All code are now available on Github at https://github.com/rachelj98/MoleratBones.